# Motivational Climate in Sport Is Associated with Life Stress Levels, Academic Performance and Physical Activity Engagement of Adolescents

**DOI:** 10.3390/ijerph16071198

**Published:** 2019-04-03

**Authors:** Manuel Castro-Sánchez, Félix Zurita-Ortega, Eduardo García-Marmol, Ramón Chacón-Cuberos

**Affiliations:** 1Department of Didactics of Musical, Plastic and Corporal Expression, University of Granada, 18071 Granada, Spain; felixzo@ugr.es; 2Department of Physical Education, University of Granada, 18071 Granada, Spain; eduardogarcia@ugr.es; 3Department of Research Methods and Diagnosis in Education, University of Granada, 18071 Granada, Spain; rchacon@ugr.es

**Keywords:** motivational climate, stress, academic performance, physical activity

## Abstract

The present study sought to define and contrast an explanatory model incorporating motivational climate towards sport, life stress, academic performance, and engagement in physical activity, and to analyze the existing relationships between these variables as a function of sex. A total of 2452 adolescents of both sexes (42.7% males and 57.3% females) participated in the present study, with self-reported ages between 13 and 16 years (M = 14.43; SD = 1.15). Participants were from Granada (Spain) and perceived motivational climate towards sport (PMCSQ-2), life stress (PSS), academic performance, and engagement in physical activity (PAQ-A) were analyzed. A multi-group structural equation model was constructed, which demonstrated excellent fit to the observed data (χ^2^ = 309.402; DF = 40; *p* < 0.001; CFI = 0.973; NFI = 0.970; IFI = 0.973; and RMSEA = 0.052). A negative and direct association exists between ego climate and task climate. A positive association was found between motivational climate, task climate (males r = 0.336/females r = 0.238), and ego climate (males r = 0.198/ females r = 0.089) and engagement in physical activity. A task climate was associated with better academic performance and lower levels of life stress. The main conclusions of this study highlight that a task-involving climate and engagement in physical activity are both associated with lower levels of life stress and higher levels of academic performance.

## 1. Introduction

In the ambit of psychology, motivation constitutes a fundamental factor in the study of human behavior. This concept provides a foundation from which human behavior can be explored and is based on the idea that individuals enact or avoid certain behaviors as a result of their motivations [1]. Within the scope of physical activity or sport engagement, the examination and analysis of motivation helps improve understanding of the diverse psychological processes related to the behaviors observed in this field [2]. Diverse factors interact within the motivational process, with these including biological factors, along with cognitive, social, and emotional factors, among others. These factors in turn determine the choice of a determined activity, the intensity with which it is engaged in, persistence, and the performance that is ultimately obtained [3].

Achievement goal theory focus on the examination of the management of motivation by individuals faced with various situations, orientating them towards the achievement of success. The theory of achievement goals is based on the idea that people execute each action intentionally, governed by several objectives associated with a determined goal in a rational manner. These are based on the expectations and values that the physical activity practitioner emphasizes to achieve success or failure [4]. Thus, the perception of success or failure is conditioned by two factors. The first is the personal characteristics of the processes of socialization, which are characterized by the previous experiences that shape the perception of achievement that each person develops. In fact, this process will influence in the conception of the own skills. An example of these premises is associated with the fact that each subject may perceive its ability in comparison with the rest, being able to find two different situations or perceptions. One subject can be perceived with the same skill level, while another one can interpret it as a high or low ability due to their personal requirements. The second is the achievement goals developed by each person. This theory influences the idea that there are two types of motivational orientation based on the concept of perceived ability [5,6].

The first motivational orientation is the ego-oriented motivation, which consists on valuing one’s own ability by comparing oneself with other people, giving priority to the result of one’s behavior over effort and execution. On the other hand, the task-oriented motivation is characterized by assessing the ability of the individual in a criterial way, prioritizing effort and execution more than the results obtained. This means that, during the sports practice, the personal improvement and a greater mastery of sports skills is pursued [5,6]. In their daily life, individuals interpret the different situations that they face, defining success and failure in an individual way as a function of the predominating motivational orientation that they adopt [7]. If an individual orients towards the task, they will focus on overcoming personal challenges, putting a special emphasis on effort and interpreting situations in which they make mistakes as an important part of progress. On the other hand, if the predominant orientation is towards the ego, the individual will be focused on bettering the performance of others and purely demonstrating their ability. As a result, any mistake is considered as failure as opposed to being an important part of personal progress [8].

Aside from the dispositional factors related with the personal motivational orientation of each individual, environmental factors also exist such as the prevailing motivational climate, which promote the diverse agents involved in the context of physical activity engagement [9]. Further, the involved agents, such as the trainers/coaches or physical education teachers can promote motivational climates, which are oriented either towards the task or the ego. A certain climate can emerge over the other as a function of the characteristics of these agents. This in turn impulses their athletes or students to become oriented predominantly towards the ego or towards the task [10]. As much in the academic context as in physical activity/sport, motivation constitutes a fundamental element in all teaching-learning processes. This can be seen through the fact that individuals who identify objectives that they seek to achieve in the form of personal goals, will also acquire a higher motivation and their learning will be more efficacious and long lasting [11]. The motivational climate is considered as a set of signals, as much explicit as implicit (use of feedback, grouping systems, type of evaluation, and task design), which predispose individuals to achieve success. Depending on how this set of signals is used, motivational climates will be created, which are oriented towards ego and performance, or towards task and mastery [12].

In the context of team sports, coaches will create motivational climates that will affect the motivational orientation of athletes, generating more frequently a task-oriented motivational climate. This strengthens the ethic work, the persistence and the improvement in the executions, also strengthening the social responsibility and the capacity of cooperation between the members of the group, as well as increasing the interest for learning [13]. The task-oriented motivational climates protect athletes from feelings of frustration and demotivation. Nevertheless, athletes who work in a context related to ego-oriented climate prevail and tend to have a greater difficulty maintaining high-perceived competence. This will increase when they overcome the opponent and demonstrate greater skill than the rest, but will decline when this objective is not achieved, thus reducing their interest in carrying out the activity and decreasing their level of effort [14].

Following the indications of Cervelló et al. [15], Duda et al. [16], and Chase [17], athletes who work in motivational contexts focused on the task show behaviors that promote maximum motivation in physical activity, regardless of the level of their skill perceptions. This is because athletes judge their skill level with their own reference standards and not with the rest of their teammates and rivals. This goal disposition implies a greater effort and persistence in the realization of task and exercises, a lower state-anxiety and a greater level of enjoyment for the accomplishment of the activities. However, motivational climates focused on the ego are characterized by focusing on the adaptive model of achievement, in which, if athletes perceive high levels of ability, they will be motivated to persist in the task. On the contrary, if the perceived ability is low, a model of little adaptive achievement will be developed, which implies a reduction of effort, high state-anxiety, attributions centered on the ability and a negative response to failure, decreasing motivation, and the persistence in the activity.

Therefore, athletes who are developed in task-oriented contexts often choose realistic roles, losing the fear of failure, regardless of showing their perceived competence. On the contrary, athletes who are developed in ego-oriented contexts try to choose tasks in which success is guaranteed or in which their disadvantage is obvious and nobody expects their victory, having a worse performance in situations in which they are evaluated [18]. The present study uses a sample of adolescents. This population is especially vulnerable to stressful situations as these individuals find themselves at a life stage during which a large number of changes occur at a physical, cognitive, emotional and social level [19]. Life stress in adolescents is a worrying factor as much for parents and teachers, as for the young people themselves. High levels of stress in the population at this age can occur due to a number of factors. Those which stand out include: Relationships with their peer group, family relationships, and situations interpreted by individuals as being threatening (exams, competitions, sentimental relationship, etc.), among others [20]. Life stress is associated with motivation. Specifically, it is found that task motivational climates are related with lower levels of life stress, while ego motivational climates are associated with higher levels of life stress, largely due to fear of failure [21].

When physical activity is engaged in on a regular basis it leads to diverse benefits at a physiological, cognitive and social level. Among the benefits derived from physical activity that stand out most are improvements in bone density, reduction of the risk of suffering chronic illnesses, reduction of anxiety levels, and improvement of social relations [22]. Further, a direct association has been found between regular engagement in physical activity and a reduction of life stress levels in those who participate [23]. When an individual engages in physical activity the production of endorphins increases, cortisol levels decrease, and general mood state improves. For this reason, it has been proposed as an alternative for reducing the life stress levels of adolescents [24].

Reviewing the existing literature, there is a positive association between the motivational climate and the practice of physical activity [18]. Specifically, an inverse relationship is observed between the prevailing ego motivational climate and the subjects’ life stress levels [25]. Moreover, there is positive associations between the task climate and academic performance, or negative associations between the ego climate and academic performance [26]. In addition, an inverse association has been found between the practice of physical activity and the life stress levels of the individuals who practice it [27], as well as a higher academic performance in regular physical activity practitioners [28]. For these reasons, we propose the structural equations model for this research that relates the mentioned factors.

Finally, the last factor to be analyzed in the present study is the academic performance of adolescents. This has been found to be influenced by multiple social factors which are related with specific characteristics of the adolescent life stage. These factors can include but are not limited to family functioning, social and sentimental relationships, life satisfaction, and self-concept [29]. Further, a direct association has been found between physical activity engagement and the academic performance of adolescents. This shows that those individuals who participate regularly in physical activity or sport tend to achieve a better level of academic performance [30].

In consideration of the evidence discussed above, the present study proposes the following objectives: 

• To define and contrast an explanatory model incorporating motivational climate towards sport, life stress, academic performance, and participation in physical activity.

• To analyze, through multi-group structural equations, the existing associations between the motivational climate towards sport, life stress, academic performance, and participation in physical activity as a function of the sex of adolescents.

## 2. Materials and Methods

### 2.1. Design and Participants

This descriptive cross-sectional study was conducted with an overall sample of 2452 adolescents of both sexes (42.7% male and 57.3% female), aged between 13 and 16 years (M = 14.43 years; SD = 1.15), who were enrolled on obligatory secondary education courses in Granada (Spain). Sample selection was conducted through a process of convenience sampling, meeting the criteria of being enrolled in obligatory secondary education in Granada, not suffering from any type of pathology that impeded participation in the study and that they practice some kind of sport out of school. The sample was obtained from nine educational centers in Granada, with all centers who voluntarily agreed to collaborate being invited to participate. 

### 2.2. Variables and Instruments

• Motivational climate (PMCSQ-2). Questionnaire extracted from the original version developed by Newton et al. [31] and adapted into Spanish by González-Cutre et al. [32]. This instrument is composed of 33 items rated along a five-point Likert scale ranging from 1 = ‘Totally disagree’ to 5 = ‘Totally agree’. The questionnaire establishes two categories: Task climate, with its dimensions of cooperative learning, effort/improvement and important role, and ego climate with its respective dimensions of punishment of mistakes, unequal recognition and rivalry between group members. Internal consistency (Cronbach’s alpha) of this instrument in its adaptation to Spanish was obtained by González-Cutre et al. [32] and was α = 0.90 for ego climate and α = 0.84 for task climate. In the present study, a value of α = 0.93 was obtained for ego climate and α = 0.92 for task climate.

• Perceived stress scale (PSS). This instrument is extracted from the original version conceived by Cohen et al. [33] and adapted into Spanish by Remor [34]. The questionnaire is composed of 14 items rated along a five-point Likert scale which ranges from 0 = ’Never’ to 4 = ’Very often’. The scale evaluates the level of life stress perceived by the respondent in the month prior to completing the questionnaire. The score obtained is inverted for items 4, 5, 6, 7, 9, 10, and 13, with all items then being summed to produce an overall score. Internal consistency (Cronbach’s alpha) of the instrument obtained by Remor [34] in its validation into Spanish was α = 0.81, with a similar value of α = 0.84 being obtained in the present study.

• Physical activity engagement (PAQ-A). The original questionnaire conceived by Kowalski et al. [35] was used in its validated version adapted into Spanish [36]. The questionnaire evaluates participation of individuals in physical activity over the seven days prior to administration of the questionnaire. It is considered to provide an indicator of adherence to physical activity and/or sport. The instrument is composed of 9 items which relate to the type and frequency of the activities engaged in. It is rated along a six-point Likert scale, which ranges from 0 = ‘Never’ to 5 = ‘Always’, producing an overall summed score whose value indicates a lower or higher frequency of physical activity engagement. Internal consistency of this instrument was obtained as α = 0.89, which is acceptable and slightly higher than the value obtained by Kowalski et al. [36] in their original study (α = 0.79).

• Academic performance. This was obtained through the calculation of each individual’s average grade over the last year and was taken from their personal file. 

### 2.3. Procedure

The Faculty of Education Sciences at the University of Granada contacted the Office of Education of the Council of Andalusia requesting collaboration from the selected Educational Centres of Granada, this providing the convenience sample. The board of directors at each educational center was informed about the nature of the study and participation of the center’s pupils was requested. Due to the participants being underage, a model of authorization destined for legal guardians was included, requesting informed consent. 

Anonymity of participants was guaranteed at all times, making it clear that collected data was to be used only for scientific purposes. Researchers were present during data collection in order to guarantee that the process was carried out correctly and to resolve any doubts. 

Following data collection, 102 questionnaires had to be discarded due to incorrect completion. The present study followed the recommendations laid out by the Declaration of Helsinki (World Medic Association, 2008) relating to research studies, in addition to national legislation regarding clinical trials (Law 223/2004 of the 6th of February), biomedical research (Law 14/2007 of the 3rd of July), and participant confidentiality (Law 15/1999 of the 13th of December). 

### 2.4. Data Analysis

For the basic descriptive analysis the statistical software program IBM SPSS^®^ (IBM Corp, Armonk, NY, USA) version 22.0 for Windows was used. The program IBM AMOS^®^ 23 (IBM Corp, Armonk, NY, USA) was employed with the purpose of analyzing the existing relationships between the constructs included in the structural model. Following development of the theoretical model a path analysis was conducted which considered the correlation matrix developed via a multi-group analysis, grouping participants according to sex. Finally, two different structural models were configured with the purpose of confirming whether the associations between the studied variables varied as a function of the sex of adolescents. 

The path models were formed by nine observable variables and two latent variables in order to determine the indicators (Figure 1). Causal explications of the latent variables are given by the proposed models via the relationships observed between the indicators, whilst also considering the reliability of measurements. Similarly, measurement error is included for the observable, variables enabling it to be directly controlled. The unidirectional arrows represent lines of influence between the latent and observable variables, with these being interpreted as multivariate regression coefficients. Bidirectional arrows show the relationship between the latent variables and are also represented by regression coefficients.

Task climate (TC) and ego climate (EC) act as exogenous variables with each one being inferred by three indicators. The indicators relating to task climate are IR (important role), E/I (effort/improvement), and CL (cooperative learning). With regards to ego climate, the indicators are PM (punishment of mistakes), UR (unequal recognition), and MR (membership rivalry). Level of participation in physical activity (PA) acts as an endogenous variable which receives the effects of the task climate (TC) and the ego climate (EC). Academic performance (AP) acts as an endogenous variable, receiving the effects of the task climate (TC), the ego climate (EC), participation in physical activity (PA), and life stress (STRESS). Likewise, life stress (STRESS) acts as an endogenous variable, receiving the effects of the task climate (TC), the ego climate (EC) and participation in physical activity (PA). 

Model fit was checked with the purpose of verifying compatibility of the model with the observed empirical information. Reliability of model fit was established according to the goodness of fit criteria proposed by Marsh [37]. 

## 3. Results

The structural equation model proposed according to sex of the analyzed pupils reveals a good fit for all evaluation indices. Chi-squared analysis produced a significant *p*-value (χ^2^ = 309.402; DF = 40; *p* < 0.001). However, it must be highlighted that this index cannot be interpreted in a standardized way and is also sensitive to changes in sample size (Marsh, 2007, p.785). In order to address these weaknesses, other indices of standardized fit were employed which are less sensitive to simple size. The value obtained for the comparative fit index (CFI) was 0.973, this being excellent. The value acquired for the normalized fit index (NFI) was 0.970 and for the incremental fit index (IFI) was 0.973, both being excellent. The root mean squared error approximation (RMSEA) obtained an adequate value of 0.052.

Both Figure 2 and Table 1 show the estimated values for the parameters of the structural model developed with adolescent males. These parameters must present an adequate magnitude and the effects should be significantly different from zero. Further, improper estimations such as negative variances should not be obtained.

The relationship between task climate and ego climate is significant at the level of *p* < 0.001, this being negative and indirect (r = −0.315).

When analyzing the influence of the indicators of motivational climate, statistically significant differences at the level of p < 0.001 were found, with all of these associations being positive and direct. In the case of task climate, the indicators show a similar type of influence, with collaborative learning being the indicator to demonstrate the greatest correlation coefficient (r = 0.896), followed by important role (r = 0.886) and effort or improvement (r = 0.882). With regards to ego climate, the indicator to exercise the greatest influence is unequal recognition (r = 0.939), followed by punishment of mistakes (r = 0.854) and group rivalry (r = 0.623).

In the same way, significant associations are observed (*p* < 0.001) in the relationships examined between motivational climate and engagement in physical activity, these being positive and direct, as much in the case of task climate (r = 0.336) as in the case of ego climate (r = 0.198). A significant association (*p* < 0.001) was also found between motivational climate and academic performance. In the case of task climate, the association found is positive and direct (r = 0.294), while in the case of ego climate this association is negative and indirect (r = −0.133).

When considering the association between motivational climate and levels of life stress, the existence of a statistically significant association at the level of *p* < 0.01 was confirmed. In the case of ego climate, the relationship shown in positive and direct (r = 0.105). In the case of task climate, the association is significant at the level of *p* < 0.001, this being negative and indirect (r = −0.300).

Engagement in physical activity and life stress reveals a negative and indirect relationship at the level of *p* < 0.001 (r = −0.160), with only a weak correlation being observed. With regards to the relationship given between engagement with physical activity and academic performance, this is positive and direct with a medium correlation strength (r = 0.265; *p* < 0.001). 

Finally, a relationship at the level of *p* < 0.01 is found between levels of life stress and academic performance, this being positive and direct (r = 0.073), and showing a weak correlation strength. 

Both Figure 3 and Table 2 show the estimated values for the parameters of the structural model developed with female adolescents. These values must present an adequate magnitude and the effects should be significantly different from zero. Further, inappropriate estimations such as negative variances must not be obtained.

The relationship between task climate and ego climate is significant at the level of *p* < 0.001, this being negative and indirect (r = −0.341).

When analyzing the influence of the indicators of motivational climate, statistically significant differences at the level of *p* < 0.001 were found, with all of these relationships being positive and direct. In the case of task climate, the indicator to exercise the greatest influence is important role (r = 0.924), followed by cooperative learning (r = 0.895), and effort or improvement (r = 0.873). In the case of ego climate, the indicator to exercise the greatest influence is unequal recognition (r = 0.948), followed by punishment of mistakes (r = 0.841), and group rivalry (r = 0.656).

In the same way, significant associations (*p* < 0.001) are observed in the relationships given between task climate and engagement in physical activity, these being positive and direct (r = 0.238). In the case of ego climate, the association found demonstrates a significance level of *p* < 0.01, with this relationship being both positive and direct (r = 0.089). A significant association is found between motivational climate and academic performance. In the case of task climate, the examined association is positive and direct (r = 0.272; *p* < 0.01), whilst in the case of ego climate, it is negative and indirect (r = −0.153; *p* < 0.001).

In the case of the association between the motivational climate and levels of life stress, the existence of a statistical association at the level of *p* < 0.001 is confirmed. In the case of ego climate an association exists which shows a positive and direct relationship (r = 0.145). With respect to a task climate, the association is negative and indirect (r = −0.121).

Engagement in physical activity and levels of life stress revealed a negative and indirect relationship at the level of *p* < 0.001 (r = −0.125), with only a weak correlation being observed. With regards to the relationship given between engagement in physical activity and academic performance, this is positive and direct with a medium association strength (r = 0.222; *p* < 0.001). 

Finally, a relationship at the level of *p* < 0.001 is found between levels of life stress and academic performance, with this being positive and direct (r = 0.095), and showing a weak correlation strength.

## 4. Discussion

The present study conducted a multi-group structural equation analysis with the aim of comparing the associations given between the motivational climate towards sport, academic performance, levels of life stress and engagement in physical activity. The path analysis developed demonstrated excellent fit indices, suggesting that a valid explanatory model had been configured, which enabled the associations between motivational and academic factors, life stress, and physical activity engagement in adolescents of both sexes to be explained. In the same way, various previous studies have been conducted, which have taken the same approach. These include studies conducted in References [38,39,40,41,42].

The structural equation model proposed shows a significant and inverse relationship between a task climate and an ego climate in both sexes, with a stronger relationship being evident amongst females. The theoretical model presented in the present study supports that an inverse association exists between a motivational climate that is oriented towards the task and one that is oriented towards the ego. In this case, adolescents that perceived a stronger task climate, tend to focus on factors related with overcoming personal challenges, effort and cooperative learning. At the same time such individuals tend to perceive lower levels of ego climate, in which they focus purely on demonstrating ability and overcoming rivals. This situation is reversed in the case of individuals who present higher levels of ego climate and therefore lower levels of task climate [43,44]. This inverse association between an ego climate and a task climate shows a greater strength of correlation within females, with males tending to report higher scores for ego climate, whilst higher scores for task climate are obtained in females. A possible explanation for this is that male adolescents at this age are often moved to a greater extent towards the achievement of extrinsic goals, motivated by competitiveness. On the other hand, females are more likely to strive for the achievement of intrinsic rewards such as improvement and personal development [45,46].

The indicator to exert the greatest influence in the case of a task-involving climate was found to be cooperative learning in males and important role in the case of females. With regards to an ego-involving climate, unequal recognition is the indicator that presents the strongest correlation, as much for males as for females. These data may be explained by the social factors related with the gender differentiation that is still present in society, such as social factors associated with gender roles [47,48,49].

The existence of a positive and direct association between the motivational climate and regular engagement in physical activity is observed, this being stronger in the case of a task climate. These relationships are stronger in males than in females. These data coincide with studies previously conducted in the adolescent context [50,51,52], with a task climate being found to be related with regular participation in physical activity. This is due to the aforementioned motivational climate being related with enjoyment when participating in physical activity and the subsequent greater adherence to physical activity or sport [40,53]. Alternatively, an ego climate is associated with greater competitiveness, causing individuals to value competition and the attainment of positive results [45].

When analyzing the relationship between motivational climate and academic performance, a positive and direct association is found between a task climate and academic performance, whilst a negative and indirect association is found with an ego climate. These data indicate that the pupils who perceived a greater task-involving climate also present better academic performance, whilst those who obtain higher scores for an ego-involving climate have inferior academic performance. These data may be explained by the characteristics of the motivational orientations of adolescents, in that individuals who are oriented towards the task focus on effort and personal betterment, working towards improvement of their abilities through training [54]. On the other hand, an ego climate is related with the pure demonstration of ability and competition, investing less importance in effort for the improvement of personal abilities [50]. For these reasons, a task climate is positively related with academic performance, while an ego climate is inversely related. 

According to the association between the motivational climate and levels of life stress, a negative relationship was found with regards to a task climate, which was evident in both males and females, though the relationship was higher in the case of males. In the case of ego climate, the relationship is positive in both sexes being higher in females. These data corroborate those found in previous studies [55,56,57], with lower levels of life stress being found in individuals focused on task-oriented climates and higher levels of life stress in those related to ego-involving climates. Thus, the adolescents more associated with task climates present less life stress levels than those related to ego-involving climate. This is due to them concentrating on improvement and personal effort, and considering mistakes as a part of learning. All of which protect against high levels of life stress [58]. Individuals who focus to a greater extent on the ego climate present higher levels of life stress, due to this type of motivational orientation being centered on the pure demonstration of abilities and overcoming rivals. This is characterized by a fear of failure [59]. 

A negative association between engagement in physical activity and life stress was observed in the adolescents analyzed, with a stronger relationship being evident amongst males. Regular engagement in physical activity is associated with the levels of life stress of participants [60,61]. This reduction in life stress occurs due to the release of endorphins which have a natural calming effect on the nervous system and reduce cortisol levels [62]. Further, engagement in physical activity has been shown to be an efficacious alternative strategy for overcoming or avoiding feelings of frustration, improving the mood state and reducing levels of anxiety and life stress [63]. Thus, from this association it is deduced that participation in regular physical activity will improve the state of psychological health of the individual, largely due to improvement in the individual’s mood state [64].

A positive association was found when analyzing the relationship between engagement with physical activity and academic performance, this being found to be stronger amongst females. These data coincide with those found by various studies, such as that conducted by Ardoy et al. [65], which analyzed the relationship between participation in physical activity and academic performance in a group of adolescents. This study demonstrated that an intervention employing a double session of intense physical activity within the examined group of adolescents, improved academic performance at a cognitive level. From these data it can be assumed that cognitive processes exist which improve in response to habitual physical activity engagement. Improvement in these processes may be evident in process resolution, memory capacity, information processing, and speed in reasoning, among others [66,67].

Finally, in analyzing the association between life stress and academic performance, a positive relationship is found which shows a greater strength of correlation amongst females. This association may be due to adolescents who exert more effort and worry more about their academic performance presenting higher levels of life stress [68]. This aforementioned result should be interpreted with caution as a high level of life stress should not be considered as leading to better academic performance. Instead those pupils who show more concern for their academic performance are liable to experience a process in which they may develop higher levels of anxiety due to a fear of failure [69]. This fear of failure may be highly influenced by the pressure suffered by some pupils at the hands of their parents or through social media [70], with this potentially leading to anxiety problems and depression [71].

The present study reports a number of limitations. Amongst these, it should be highlighted that the descriptive and cross-sectional study design does not allow causal relations to be established. For these reasons, the results obtained in the present investigation should be interpreted with caution. In addition, it would have been interesting to include a larger number of psychological variables in the study with the aim of analyzing their relationship with the factors analyzed in the present study. As has been previously indicated, it was not possible to establish precise causes of the associations between the analyzed variables. The outcomes of the present study suggest that a practical implication could be to strengthen engagement with physical activity through physical education classes, as this is associated with better academic performance and lower levels of life stress in adolescents. This will lead to improved quality of life. In addition, it would be interesting to create motivational climates oriented towards the task, due to the greater benefits of these with respect to participation in playful physical activity.

## 5. Conclusions

As the main conclusion of the present study a greater task-climate relative to the ego was found to exist, with an inverse relationship being found between both motivational climates, this being stronger amongst females. A positive association exists between the motivational climate and engagement with physical activity, with this being stronger in the case of a task climate and in male adolescents. The task climate is directly related with better academic performance, whilst an ego climate is associated with worse academic performance. A positive association exists between an ego climate and levels of life stress in adolescents, and a negative association in the case of a task climate. When the ego climate increases, so does the level of life stress, while, when the task-involving climate increases, life stress decreases. Greater adherence to physical activity is associated with a lower level of life stress and an improvement in academic performance. Finally, a positive association is found between life stress and academic performance, with better academic performance being evident among adolescents who present greater levels of life stress.

## Figures and Tables

**Figure 1 ijerph-16-01198-f001:**
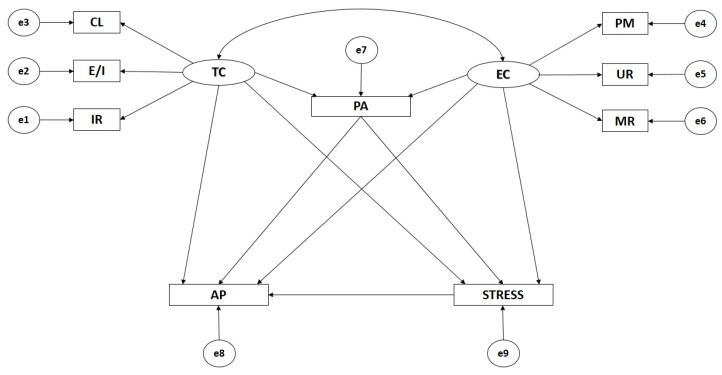
Theoretical Model. Note: TC, task climate; CL, cooperative learning; E/I, effort/improvement; IR, important role; EC, ego climate; MR, member rivalry; PM, punishment of mistakes; UR, unequal recognition; PA, physical activity; AP, academic performance; and STRESS, life stress.

**Figure 2 ijerph-16-01198-f002:**
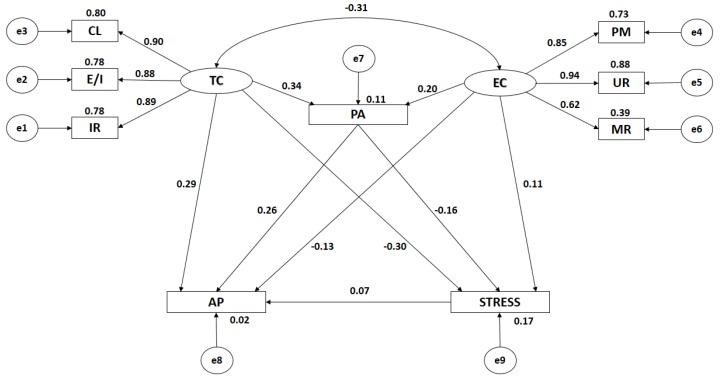
Structural equation model incorporating only males. Note: TC, task climate; CL, cooperative learning; E/I, effort/improvement; IR, important role; EC, ego climate; MR, member rivalry; PM, punishment of mistakes; UR, unequal recognition; PA, physical activity; AP, academic performance; and STRESS, life stress.

**Figure 3 ijerph-16-01198-f003:**
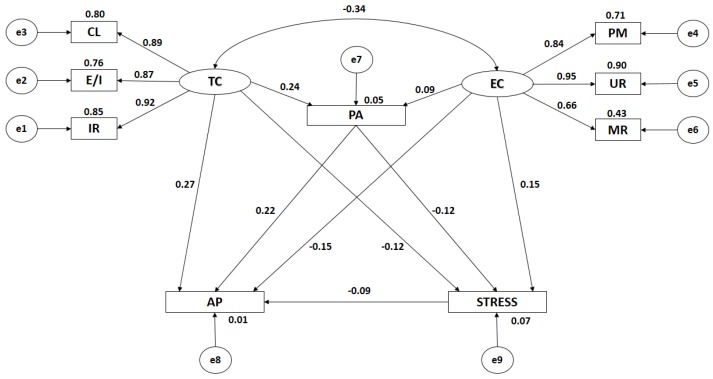
Structural equation model for females only. Note: TC, task climate; CL, cooperative learning; E/I, effort/improvement; IR, important role; EC, ego climate; MR, member rivalry; PM, punishment of mistakes; UR, unequal recognition; PA, physical activity; AP, academic performance; and STRESS, life stress.

**Table 1 ijerph-16-01198-t001:** Structural model for males.

Associations between Variables			R.W.				S.R.W.
			Estimations	S.E.	C.R.	*p*	Estimations
PA	←	TC	0.392	0.039	10.158	***	0.336
PA	←	EC	0.227	0.038	6.029	***	0.198
STRESS	←	EC	1.232	0.378	3.264	**	0.105
STRESS	←	TC	−3.580	0.402	−8.897	***	−0.300
STRESS	←	PA	−1.630	0.309	−5.274	***	−0.160
IR	←	TC	1.000	−	−	***	0.886
E/I	←	TC	0.827	0.021	38.971	***	0.882
CL	←	TC	0.994	0.025	39.895	***	0.896
PM	←	EC	1.000	−	−	***	0.854
UR	←	EC	1.259	0.041	30.469	***	0.939
MR	←	EC	0.803	0.037	21.838	***	0.623
AP	←	TC	0.301	0.044	6.880	***	0.294
AP	←	EC	−1.406	0.445	−3.159	***	−0.133
AP	←	PA	0.379	0.063	6.063	***	0.265
AP	←	STRESS	8.156	3.779	2.158	**	0.073
EC	↔	TC	−0.145	0.017	−8.692	***	−0.315

Note: * *p* < 0.05; ** *p* < 0.01; *** *p* < 0.001. TC, task climate; CL, cooperative learning; E/I, effort/improvement; IR, important role; EC, ego climate; MR, member rivalry; PM, punishment of mistakes; UR, unequal recognition; PA, physical activity; AP, academic performance; STRESS, life stress. R.W., regression weight; S.R.W., standardised regression weight; S.E., standard error; and C.R., critical ratio.

**Table 2 ijerph-16-01198-t002:** Structural model for females.

Associations between Variables			R.W.				S.R.W.
			Estimations	S.E.	C.R.	*p*	Estimations
PA	←	TC	0.247	0.030	8.166	***	0.238
PA	←	EC	0.108	0.036	3.054	**	0.089
STRESS	←	EC	1.771	0.355	4.993	***	0.145
STRESS	←	TC	−1.257	0.308	−4.084	***	−0.121
STRESS	←	PA	−1.252	0.266	−4.704	***	−0.125
IR	←	TC	1.000	−	−	***	0.924
E/I	←	TC	0.776	0.016	48.363	***	0.873
CL	←	TC	0.969	0.019	50.708	***	0.895
PM	←	EC	1.000	−	−	***	0.841
UR	←	EC	1.375	0.038	35.885	***	0.948
MR	←	EC	0.937	0.035	27.022	***	0.656
AP	←	TC	0.182	0.029	6.274	***	0.272
AP	←	EC	−24.550	8.539	−2.875	**	−0.153
AP	←	PA	0.184	0.035	5.291	***	0.222
AP	←	STRESS	−1.039	0.373	−2.787	***	−0.095
EC	↔	TC	−0.176	0.016	−10.916	***	−0.341

Note: ** *p* < 0.01; *** *p* < 0.001. TC, task climate; CL, cooperative learning; E/I, effort/improvement; IR, important role; EC, ego climate; MR, member rivalry; PM, punishment of mistakes; UR, unequal recognition; PA, physical activity; AP, academic performance; STRESS, life stress. R.W., regression weight; S.R.W., standardised regression weight; S.E., standard error; and C.R., critical ratio.

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
