# Peer review of "Motivational Climate in Sport Is Associated with Life Stress Levels, Academic Performance and Physical Activity Engagement of Adolescents"

_ijerph, 2019, doi:10.3390/ijerph16071198_

Round 1
Reviewer 1 Report
The purpose of the current investigation was to examine the impact of the perceived motivational climate (in sport?) on adolescent engagement in physical activity, stress, and academic performance. A secondary purpose was to examine more nuanced relationships between these variables and whether they varied by gender. An examination of the impact of the perceived motivational climate on sport teams on these particular variables would certainly add much to the literature. Moreover, the proposed model is logical, given the achievement goal theory literature. However, as written, it is unclear the authors understand achievement goal theory or that they collected data on the perceived motivational climate worth using. For example, the authors aren’t clear as to whether the perceived motivational climate was reflective of the students’ sports teams. It seems the authors sought to examine the “motivational climate towards sport”, but this doesn’t make any sense theoretically. The motivational climate is context specific. Please provide greater clarity about the instructions participants were asked to follow.
Moreover, the authors seem to confuse goal orientations and motivational climates throughout the paper, and provide an incorrect explanation of motivational climates. There is also a high level of speculation in the discussion. I am also concerned that the authors improperly cited research throughout the manuscript. For example, a seemingly random research article was cited instead of referencing theoretical resource when attempting to explain the theoretical foundation for the study (e.g., citation #7). This is just one of many improper citations.
I also have a number of questions about the manner in which the data was prepared for analysis (e.g., were the items parceled? What were the item loadings?) that I would like addressed in future revisions. Lastly, the authors do not provide an adequate justification for their proposed model, and there are typos throughout the reference section.
The model is theoretically sound, and the findings are in line with previous research. Moreover, this would add much to the literature. I recommend the authors review the theory and provide a better explanation of achievement goal theory and a justification for the proposed model in the manuscript. And, when discussing the results, temper the language and overgeneralizations/speculation (e.g., pg. 9, lines 314-326).
Author Response
Dear the editor and reviewers,
We would like to express our gratitude for the time taken to review this manuscript and for the comments made, which we believe to be critical for producing rigorous and quality research. We have detailed below the changes made in the original article: “Motivational climate towards sport is associated with stress levels, academic performance and physical activity engagement of adolescents” (ijerph-463109).
Modifications have been made in the original manuscript following the reviewers’ comments. For each modification we have written: the original comment as written by the reviewer in addition to the page and line number; and the change made in response to that comment. Changes have been made using the tool “Track changes” enabling editor and reviewers to identify modifications easily.
Reviewer 1
Comment 1:
The purpose of the current investigation was to examine the impact of the perceived motivational climate (in sport?) on adolescent engagement in physical activity, stress, and academic performance. A secondary purpose was to examine more nuanced relationships between these variables and whether they varied by gender. An examination of the impact of the perceived motivational climate on sport teams on these particular variables would certainly add much to the literature. Moreover, the proposed model is logical, given the achievement goal theory literature. However, as written, it is unclear the authors understand achievement goal theory or that they collected data on the perceived motivational climate worth using. For example, the authors aren’t clear as to whether the perceived motivational climate was reflective of the students’ sports teams. It seems the authors sought to examine the “motivational climate towards sport”, but this doesn’t make any sense theoretically. The motivational climate is context specific. Please provide greater clarity about the instructions participants were asked to follow.
Response 1:
We appreciate the suggestions made since they will improve the quality of the manuscript. In this sense, it has proceeded to modify the title of the article by changing “toward” by “in”, because it makes more sense and the title is more accurate.
Regarding the impact of the motivational climate on sports, a brief description of how the motivational climate influences team sports has been included in the introduction in a general way. It is due to this investigation did not studied clubs. in those who practice physical activity-sports.
The motivational climate analysed in this research is referred to the sports activity practiced by adolescents after school, this being an inclusion criterion for participation in the present investigation. Data that has been included in the selection criteria of the sample, being it clearer. Thank you very much for this indication, as this shows the objective of the study with greater clarity.
Comment 2:
Moreover, the authors seem to confuse goal orientations and motivational climates throughout the paper, and provide an incorrect explanation of motivational climates. There is also a high level of speculation in the discussion. I am also concerned that the authors improperly cited research throughout the manuscript. For example, a seemingly random research article was cited instead of referencing theoretical resource when attempting to explain the theoretical foundation for the study (e.g., citation #7). This is just one of many improper citations.
Response 2:
Thank you for your indication, as it will improve the quality of the manuscript. The entire document has been revised and the wording has been modified, focusing on the analysis of the motivational climate created by the trainer and not the orientations of the adolescents. However, in some cases we talk about motivational orientations, because when the coach promotes a motivational climate, this encourages athletes to adopt a similar motivational orientation, as is done in the following investigation:
-Gjesdal, S.; Stenling, A.; Solstad, B.E.; Ommundsen, Y. A study of coach‐team perceptual distance concerning the coach‐created motivational climate in youth sport. Scand J Med Sci Spor, 2019, 29, 132-143. doi: 10.1111/sms.13306.
Regarding the references, the number 7 is used to indicate that the subjects interpret the situations of success and failure personally. It has proceeded to review the article cited, in which that explanation is provided. In any case, those that were erroneous have been revised and replaced. If you find an erroneous appointment, please inform us so we can review it and correct the errors.
Thanks for the indications.
Comment 3:
I also have a number of questions about the manner in which the data was prepared for analysis (e.g., were the items parceled? What were the item loadings?) that I would like addressed in future revisions. Lastly, the authors do not provide an adequate justification for their proposed model, and there are typos throughout the reference section.
Response 3:
Thanks for this indication. Answering your question, in the present investigation an exploratory factor analysis has not been carried out. This is because there are already validations of this instrument in many populations, in which the psychometric properties of the scale are analyzed:
-Zurita Ortega, F., Castro Sánchez, M., Chacón Cuberos, R., Cachón Zagalaz, J., Cofré Bolados, C., Knox, E., & Muros, J. (2018). Analysis of the Psychometric Properties of Perceived Motivational Climate in Sport Questionnaire and Its Relationship to Physical Activity and Gender Using Structural Equation Modelling. Sustainability, 10(3), 632.
In the present research, a theoretical model has been made as in the following investigations:
- Jaakkola, T., Ntoumanis, N., & Liukkonen, J. (2016). Motivational climate, goal orientation, perceived sport ability, and enjoyment within Finnish junior ice hockey players. Scandinavian journal of medicine & science in sports, 26(1), 109-115.
- Ruiz, M. C., Haapanen, S., Tolvanen, A., Robazza, C., & Duda, J. L. (2017). Predicting athletes’ functional and dysfunctional emotions: The role of the motivational climate and motivation regulations. Journal of sports sciences, 35(16), 1598-1606.
- Baena-Extremera, A., Gómez-López, M., Granero-Gallegos, A., & Ortiz-Camacho, M. D. M. (2015). Predicting satisfaction in physical education from motivational climate and self-determined motivation. Journal of Teaching in Physical Education, 34(2), 210-224.
- Zurita Ortega, F., Castro Sánchez, M., González, Á., Ignacio, J., Rodríguez Fernández, S., & Pérez Cortés, A. J. (2016). Autoconcepto, Actividad física y Familia: Análisis de un modelo de ecuaciones estructurales. Revista de psicología del deporte, 25(1), 0097-104.
- Cuberos, R. C., Ortega, F. Z., Sánchez, M. C., Garcés, T. E., Martínez, A. M., & Ruiz, G. R. R. (2017). Relación entre autoconcepto, consumo de sustancias y uso problemático de videojuegos en universitarios: un modelo de ecuaciones estructurales. Adicciones.
- Martínez, A. M., Sánchez, M. C., Fernández, S. R., Ortega, F. Z., Cuberos, R. C., & Garcés, T. E. (2018). Conducta violenta, victimización, autoestima y actividad física de adolescentes españoles en función del lugar de residencia: un modelo de ecuaciones estructurales. Revista de Psicología Social, 33(1), 125-141.
- Zurita-Ortega, F., Castro-Sánchez, M., Rodríguez-Fernández, S., Cofré-Boladós, C., Chacón-Cuberos, R., Martínez-Martínez, A., & Muros-Molina, J. J. (2017). Actividad física, obesidad y autoestima en escolares chilenos: Análisis mediante ecuaciones estructurales. Revista médica de Chile, 145(3), 299-308.
- Baena-Extremera, A., Granero-Gallegos, A., Ponce-de-León-Elizondo, A., Sanz-Arazuri, E., Valdemoros-San-Emeterio, M. D. L., & Martínez-Molina, M. (2016). Factores psicológicos relacionados con las clases de educación física como predictores de la intención de la práctica de actividad física en el tiempo libre en estudiantes. Ciência & Saúde Coletiva, 21, 1105-1112.
The load of each item has not been analyzed because the dimensions of the questionnaire are used for the model to comply with the principle of parsimony. However, if you think this is necessary, the authors do not mind adding it.
Thank you for your comment regarding the inclusion of an adequate justification for the proposed model. This is included at the end of the introduction, before the objectives, providing a justification for the proposed model.
The bibliography has been revised and typographical errors detected have been corrected.
Comment 4:
The model is theoretically sound, and the findings are in line with previous research. Moreover, this would add much to the literature. I recommend the authors review the theory and provide a better explanation of achievement goal theory and a justification for the proposed model in the manuscript. And, when discussing the results, temper the language and overgeneralizations/speculation (e.g., pg. 9, lines 314-326).
Response 4:
Thank you very much for your comments. We appreciate the contributions made that will improve the quality of the manuscript. A more extensive and profound explanation of the theory of achievement goals and a justification of the model proposed in the manuscript has been provided, reviewing the existing theory in order to explain the relations proposed in the theoretical model.
In the discussion section the language used has been moderated so that it is not excessively speculative, trying to stick to the results obtained. In addition, it has been added that the results must be interpreted with caution in the limitations section, since a study of these characteristics does not allow establishing cause-effect relationships.
Reviewer 2 Report
Overall comments:
Interesting and well written paper, with a large study sample. I am not sure I follow the headline Motivational climate towards sport. Could in sport be a better fit? The authors have chosen to separate female and male participants in their analysis. Have you made an argument as to why? I am furthermore not sure you have to include self-determination theory in your paper, when you don´t use the theory in your study. CL, E/I and IR are included in Task and PM, UR and MR are included in Ego. Why are climate not only described as ego and task, why are the dimensions analyzed both separately and as one dimension (normally) in the study. Could you please argue as to why? Implication of the results could be highlighted in the conclusion.
Major changes:
- In the abstract page 1, line 19, the stress measurement/questionnaire is entitled ABAE-10, while in the materials and methods it is entitled PSS? Which one f them were used?
- The results are in parts of the paper describes a bit overall. In example page 1, line 23-24: A positive association was found between motivational climate and engagement… Is the positive association between motivational climate overall or related to task or ego? It´s related to ego or task isn’t it?
- Page 3, line 141-142, it seems like the variable Academic performance is shortly described. The only line ends with .. is there any text missing? Thus, this include all courses?
- A few places in the text the results are stated in a general fashion and then made explicit in the next sentence. In example page 6. Line 219-222. Is the first sentence Statistically .. necessary when the next sentence is more precise regarding the results? Another example is page 6, line 232-233.
Minor changes:
- Page 5, line 195. Double space between indices. Chi..
Author Response
Dear the editor and reviewers,
We would like to express our gratitude for the time taken to review this manuscript and for the comments made, which we believe to be critical for producing rigorous and quality research. We have detailed below the changes made in the original article: “Motivational climate towards sport is associated with stress levels, academic performance and physical activity engagement of adolescents” (ijerph-463109).
Modifications have been made in the original manuscript following the reviewers’ comments. For each modification we have written: the original comment as written by the reviewer in addition to the page and line number; and the change made in response to that comment. Changes have been made using the tool “Track changes” enabling editor and reviewers to identify modifications easily.
Reviewer 2
Comment 1:
Interesting and well written paper, with a large study sample. I am not sure I follow the headline Motivational climate towards sport. Could in sport be a better fit? The authors have chosen to separate female and male participants in their analysis. Have you made an argument as to why? I am furthermore not sure you have to include self-determination theory in your paper, when you don´t use the theory in your study. CL, E/I and IR are included in Task and PM, UR and MR are included in Ego. Why are climate not only described as ego and task, why are the dimensions analyzed both separately and as one dimension (normally) in the study. Could you please argue as to why? Implication of the results could be highlighted in the conclusion.
Response 1:
We appreciate your indications as they improve the quality of the article. We have proceeded to correct the title of the item by indicating changing “towads” by “in”. We believe that with this title the objective of the investigation is clearer.
Regarding the separation of participants according to gender, this has been done based on previous research that indicates that there are differences in the motivational climate between men and women. In this way, a model of multigroup structural equations has been made according to sex in order to control the effect that sex could cause. In addition, references are added to various studies that qualify the existing differences in the motivational climate between men and women, such as the following:
-Breiger, J., Cumming, S. P., Smith, R. E., & Smoll, F. (2015). Winning, motivational climate, and young athletes' competitive experiences: Some notable sex differences. International Journal of Sports Science & Coaching, 10(2-3), 395-411.
-Jaakkola, T., Wang, C. J., Soini, M., & Liukkonen, J. (2015). Students’ perceptions of motivational climate and enjoyment in Finnish physical education: A latent profile analysis. Journal of sports science & medicine, 14(3), 477.
-Hogue, C. M., Fry, M. D., & Fry, A. C. (2017). The differential impact of motivational climate on adolescents’ psychological and physiological stress responses. Psychology of Sport and Exercise, 30, 118-127.
-Atkins, M. R., Johnson, D. M., Force, E. C., & Petrie, T. A. (2015). Peers, parents, and coaches, oh my! The relation of the motivational climate to boys' intention to continue in sport. Psychology of Sport and Exercise, 16, 170-180.
-Chacón, R. C., Muros, J. M., Cachón, J. Z., Zagalaz, M. S., Castro, M. S., & Zurita, F. O. (2018). Physical activity, Mediterranean diet, maximal oxygen uptake and motivational climate towards sports in schoolchildren from the province of Granada: a structural equation model. Nutricion hospitalaria, 35(4), 774-781.
-Cuberos, R. C., Ortega, F. Z., Zagalaz, J. C., Garcés, T. E., Sánchez, M. C., & Perez Cortes, A. J. (2018). Perceived Motivational Climate Toward Sport in University Physical Education Students. Apunts: Educació Física i Esports, (131).
-Chacón-Cuberos, R., Badicu, G., Zurita-Ortega, F., & Castro-Sánchez, M. (2019). Mediterranean Diet and Motivation in Sport: A Comparative Study Between University Students from Spain and Romania. Nutrients, 11(1), 30.
-Castro-Sánchez, M., Zurita-Ortega, F., Ubago-Jiménez, J., Ramírez-Granizo, I., & Chacón-Cuberos, R. (2018). Motivational Climate in Youth Football Players. Behavioral Sciences, 8(9), 83.
Regarding the theory of self-determination, the reviewer is right to mention that this theory is not used in this investigation. However, we would like to point out that it is only mentioned as one of the most used motivational theories in the field of physical activity and sport. However, if you consider it appropriate, we can eliminate the mention of motivational theory from the text.
Considering the doubt about the dimensions that make up the motivational climate in the SEM, the model constructs the exogenous variables (task climate and ego climate) based on the three indicators that make up each of the dimensions, not establishing relationships between the indicators and the rest of the variables analyzed in the present research. Therefore, the relationships provided by the model are associated with the variables “ego climate” and “task climate”, and not indicators. Thank you for your comment.
Regarding the practical implications derived from the results, they are included in the last paragraph of the discussion after the limitations. Thus, it is indicated that although they cannot establish precise causal relationship between the analyzed variables. The outcomes of the present study suggest that a practical implication could be to strengthen engagement with physical activity through physical education classes, as this is associated with better academic performance and lower levels of stress in adolescents. This will lead to improved quality of life. In addition, it would be interesting to create motivational climates oriented toward the task, due to the greater benefits of these with respect to participation in playful physical activity
Comment 2:
-In the abstract page 1, line 19, the stress measurement/questionnaire is entitled ABAE-10, while in the materials and methods it is entitled PSS? Which one f them were used?
Response 2:
The instrument used in this research is the PSS (Perceived Stress Scale). It has been a typographical error and has been modified in the abstract, indicating that stress levels have been analyzed using the PSS instrument.
Comment 3:
-The results are in parts of the paper describes a bit overall. In example page 1, line 23-24: A positive association was found between motivational climate and engagement… Is the positive association between motivational climate overall or related to task or ego? It´s related to ego or task isn’t it?
Response 3:
Thanks for this indication. It has been indicated that the positive association between the motivational climate and the adherence to the practice of physical activity occurs in both dimensions. Thanks for this indication, we believe that it will improve the understanding of the research.
Comment 4:
-Page 3, line 141-142, it seems like the variable Academic performance is shortly described. The only line ends with .. is there any text missing? Thus, this include all courses?
Response 4:
Thanks for this suggestion. The variable “academic performance” is complete, because it is described briefly as it is collected by calculating the average mark obtained in the student's academic record of the last academic year.
Comment 5:
-A few places in the text the results are stated in a general fashion and then made explicit in the next sentence. In example page 6. Line 219-222. Is the first sentence Statistically .. necessary when the next sentence is more precise regarding the results? Another example is page 6, line 232-233.
Response 5:
Thanks for this suggestion of improvement. It is true that some phrases were reiterative. Repeated sentences have been eliminated in order to improve the understanding of the manuscript.
Comment 6:
-Page 5, line 195. Double space between indices. Chi.
Response 6:
Thanks for notifying us of the error. The error has been corrected, eliminating the existing double space.
Round 2
Reviewer 1 Report
As noted in my initial review, I believe examining these variables in this particular way will add much to the literature. A notable strength of this study is the population under investigation and the researchers’ access to their grades. Arguably the most important finding in this study is the relationship between the perceived motivational climate on “out of school sport” and the life stress of the athletes. It would add greater clarity if the authors would note that this is “life stress” when referring to the relationship throughout the manuscript, including the abstract (as there are other forms of stress). I have noted my concerns below and provided a few guiding questions/points. If these concerns and the concerns of the editor/other reviewers are addressed, I would gladly provide a more review of the updated paper.
Major points:
I continue to have concerns with the study conceptualization, the authors’ understanding of achievement goal theory, and the flow of the introduction is a difficult to follow.
It is still unclear whether the authors fully grasp achievement goal theory. To begin, there is a difference between task goal orientations and task-involving motivational climates. The justification for the proposed model is centered on the role of goal orientations, which were not measured in the current investigation. Moreover, there is more to creating a task-involving motivational climates than simply “focusing on the task” (as suggested on p. 2, line 94). Also, people do not “demonstrate” climates (p. 9, line 344) – they perceive climates. These are just a few examples to highlight this particular concern.
Introduction
It is unclear how goal orientations and self-determination theory are relevant to the current investigation. A brief mention of goal orientations may be relevant in the discussion of the results, but it simply confuses readers when introduced (and explained at length) in the beginning of the paper. It may help to better explain how motivational climates are defined in the introduction, to present a rationale for the model, and to highlight the importance of this CURRENT investigation. Likewise, a more thorough review of the relevant climate literature is warranted.
The authors’ focus on goal orientation leads me to ask the question, why weren’t goal orientations examined? Much of the rationale presented in the introduction suggests goal orientations would mediate these relationships and should be included in the model.
Finally, there are important differences between state stress and life stress. The authors seem to conflate the two at times, yet these constructs should be clearly defined and the argument suggesting a relationship should be explained in detail with supporting research (see Hogue, Fry, Iwasaki 2018).
· Hogue, C. M., Fry, M. D., & Iwasaki, S. The Impact of the Perceived Motivational Climate in Physical Education Classes on Adolescent Greater Life Stress, Coping Appraisals, and Experience of Shame. Sport, Exercise, and Performance Psychology. http://dx.doi.org/10.1037/spy0000153
Minor points:
· Lines 25 & 27, add “life” before “stress”
· Because there is a notable difference in the strength of the relationship between task- and ego-involving climate perceptions and engagement in physical activity, I recommend the authors include these values in the abstract (line 24).
· Lines 41-44 It seems unnecessary to mention self-determination theory, given that the authors do not reference it again. Also, Nicholls 1984 & 1989 are typically cited together for AGT
o Nicholls, J. (1984). Achievement motivation: Conceptions of ability, subjective experience, task choice, and performance. Psychological Review, 91, 328–346. http://dx.doi.org/10.1037/0033-295X .91.3.328
· Pg 2, line 52 – replace “creates” with “emphasizes” for accuracy
· While the inclusion of additional information explaining achievement goal theory will add important information to the manuscript, the theory is a bit confusing as explained. For example…
o What do the authors mean by “personal characteristics of the processes of socialization” and “conception of the skill assumed by each person”? Please clarify for reader understanding
· Likewise, the explanation of a task-orientation should note that it is personal improvement/advancing toward skill mastery rather than simply execution
· The motivational climate is most often referred to as “task-involving” and “ego-involving” rather than task-oriented. For instance, Newton et al (2000) in developing the PMCSQ-2 referred to the climates in this way. I recommend the authors consider revising, as individuals unfamiliar with the AGT literature often confuse task-(goal) orientations with task-involving motivational climates.
· It is recommended the authors change “confirms” to “supports” p. 9, lines 344
Author Response
Dear the editor and reviewers,
We would like to express our gratitude for the time taken to review this manuscript and for the comments made, which we believe to be critical for producing rigorous and quality research. We have detailed below the changes made in the original article: “Motivational climate towards sport is associated with stress levels, academic performance and physical activity engagement of adolescents” (ijerph-463109).
Modifications have been made in the original manuscript following the reviewers’ comments. For each modification we have written: the original comment as written by the reviewer in addition to the page and line number; and the change made in response to that comment. Changes have been made using the tool “Track changes” enabling editor and reviewers to identify modifications easily.
Reviewer 1
Comment 1:
As noted in my initial review, I believe examining these variables in this particular way will add much to the literature. A notable strength of this study is the population under investigation and the researchers’ access to their grades. Arguably the most important finding in this study is the relationship between the perceived motivational climate on “out of school sport” and the life stress of the athletes. It would add greater clarity if the authors would note that this is “life stress” when referring to the relationship throughout the manuscript, including the abstract (as there are other forms of stress). I have noted my concerns below and provided a few guiding questions/points. If these concerns and the concerns of the editor/other reviewers are addressed, I would gladly provide a more review of the updated paper.
Response 1:
We appreciate the suggestions made since they will improve the quality of the manuscript. In this sense, we have indicated throughout the manuscript that the concept is “life stress”.
Comment 2:
Major points:
I continue to have concerns with the study conceptualization, the authors’ understanding of achievement goal theory, and the flow of the introduction is a difficult to follow.
It is still unclear whether the authors fully grasp achievement goal theory. To begin, there is a difference between task goal orientations and task-involving motivational climates. The justification for the proposed model is centered on the role of goal orientations, which were not measured in the current investigation. Moreover, there is more to creating a task-involving motivational climates than simply “focusing on the task” (as suggested on p. 2, line 94). Also, people do not “demonstrate” climates (p. 9, line 344) – they perceive climates. These are just a few examples to highlight this particular concern.
Response 2:
Thank you for your indication, as it will improve the quality of the manuscript. The proposed modifications have been made in order to comply with the instructions of the reviewer.
Thanks for the indications.
Comment 3:
Introduction
It is unclear how goal orientations and self-determination theory are relevant to the current investigation. A brief mention of goal orientations may be relevant in the discussion of the results, but it simply confuses readers when introduced (and explained at length) in the beginning of the paper. It may help to better explain how motivational climates are defined in the introduction, to present a rationale for the model, and to highlight the importance of this CURRENT investigation. Likewise, a more thorough review of the relevant climate literature is warranted.
The authors’ focus on goal orientation leads me to ask the question, why weren’t goal orientations examined? Much of the rationale presented in the introduction suggests goal orientations would mediate these relationships and should be included in the model.
Finally, there are important differences between state stress and life stress. The authors seem to conflate the two at times, yet these constructs should be clearly defined and the argument suggesting a relationship should be explained in detail with supporting research (see Hogue, Fry, Iwasaki 2018).
Hogue, C. M., Fry, M. D., & Iwasaki, S. The Impact of the Perceived Motivational Climate in Physical Education Classes on Adolescent Greater Life Stress, Coping Appraisals, and Experience of Shame. Sport, Exercise, and Performance Psychology. http://dx.doi.org/10.1037/spy0000153.
Response 3:
Thanks for this indication. The introduction has been modified to make it clearer.
In terms of stress levels, it has been indicated throughout the document that it refers to the life stress, rather than to a stress state.
Comment 4:
·Lines 25 & 27, add “life” before “stress”
Response 4:
Thank you very much for your comment. We appreciate the contributions made that will improve the quality of the manuscript. In lines 25 & 27, we add “life” before “stress”.
Comment 5:
·Because there is a notable difference in the strength of the relationship between task- and ego-involving climate perceptions and engagement in physical activity, I recommend the authors include these values in the abstract (line 24).
Response 5:
Thank you very much for your comments. We have included these values in the abstract (line 24).
Comment 6:
Lines 41-44 It seems unnecessary to mention self-determination theory, given that the authors do not reference it again. Also, Nicholls 1984 & 1989 are typically cited together for AGT
Nicholls, J. (1984). Achievement motivation: Conceptions of ability, subjective experience, task choice, and performance. Psychological Review, 91, 328–346. http://dx.doi.org/10.1037/0033-295X .91.3.328
Response 6:
Thank you very much for your comments. The paragraph in which we talk about the two motivational theories has been eliminated. We have directly included the theory of achievement goals, adding the reference of Nicholls (1984) to explain this motivational theory.
Comment 7:
·Pg 2, line 52 – replace “creates” with “emphasizes” for accuracy
Response 7:
Thank you very much for your comment. In Pg 2, line 52 – we have replaced “creates” with “emphasizes” for accuracy.
Comment 8:
While the inclusion of additional information explaining achievement goal theory will add important information to the manuscript, the theory is a bit confusing as explained. For example…
What do the authors mean by “personal characteristics of the processes of socialization” and “conception of the skill assumed by each person”? Please clarify for reader understanding
Response 8:
Thank you very much for your comments. We appreciate the contributions made that will improve the quality of the manuscript. A brief explanation of these two sentences has been added in order to to clarify its meaning.
Comment 9:
·Likewise, the explanation of a task-orientation should note that it is personal improvement/advancing toward skill mastery rather than simply execution
Response 9:
Thank you very much for your comment. We believe that its clarification improves the understanding of the manuscript. It has been specified that the orientation to the task refers to a personal improvement and to the progress towards the mastery of the task, and not only towards a simple execution of the task.
Comment 10:
·The motivational climate is most often referred to as “task-involving” and “ego-involving” rather than task-oriented. For instance, Newton et al (2000) in developing the PMCSQ-2 referred to the climates in this way. I recommend the authors consider revising, as individuals unfamiliar with the AGT literature often confuse task-(goal) orientations with task-involving motivational climates.
Response 10:
Thank you very much for your comment. The literature has been revised and errors have been corrected. Thank you very much for your support.
Comment 11:
·It is recommended the authors change “confirms” to “supports” p. 9, lines 344
Response 11:
Thank you very much for your comment. We appreciate the contributions made that will improve the quality of the manuscript. The authors have changed “confirms” to “supports” p. 9, lines 344.
Reviewer 2 Report
non- the comments have been answered in a satisfying matter.
Author Response
Thank you very much for your comments.